# Broadband giant-refractive-index material based on mesoscopic space-filling curves

Taeyong Chang[1], Jong Uk Kim[1], Seung Kyu Kang[1], Hyowook Kim[1], Do Kyung Kim[1], Yong-Hee Lee[2] & Jonghwa Shin[1]

The refractive index is the fundamental property of all optical materials and dictates Snell's law, propagation speed, wavelength, diffraction, energy density, absorption and emission of light in materials. Experimentally realized broadband refractive indices remain <40, even with intricately designed artificial media. Herein, we demonstrate a measured index >1,800 resulting from a mesoscopic crystal with a dielectric constant greater than three million. This gigantic enhancement effect originates from the space-filling curve concept from mathematics. The principle is inherently very broad band, the enhancement being nearly constant from zero up to the frequency of interest. This broadband giant-refractive-index medium promises not only enhanced resolution in imaging and raised fundamental absorption limits in solar energy devices, but also compact, power-efficient components for optical communication and increased performance in many other applications.

[1] Department of Materials Science and Engineering, KAIST, Daejeon 34141, Republic of Korea. [2] Department of Physics, KAIST, Daejeon 34141, Republic of Korea. Correspondence and requests for materials should be addressed to Y.-H.L. (email: yhlee@kaist.ac.kr) or to J.S. (email: qubit@kaist.ac.kr).

There exists a fundamental upper bound on the refractive index of any natural or artificial medium with atomic scale unit cells. For non-magnetic materials, the refractive index ($n$) is solely determined by the dielectric constant ($\varepsilon_r$), which in turn is determined by the atomic (molecular) polarizability and its spatial arrangement. The volume-averaged polarizability of an ensemble of ideal two-level systems is summarized by the factor $ND^2(\omega_t - \omega)/\varepsilon_o\hbar[(\omega_t - \omega)^2 + \gamma^2]$, where $N$ is the number density of the two-level system, $D$ is the relevant transition dipole moment, $\hbar$ is the reduced Planck constant, $\omega_t$ is the transition frequency between two levels and $\gamma$ is the effective damping factor[1]. For low frequencies ($\omega \ll \omega_t$), this reduces to $ND^2/\varepsilon_o\hbar\omega_t$. For typical $N$ and $D$ of solids, $ND^2/\varepsilon_o\hbar\omega_t$ is on the order of unity, which is why the refractive indices of materials remain also on the order of unity. If one can increase this factor by six orders of magnitude, the dielectric constant would increase by the same amount and the refractive index, by three orders.

Existing approaches to increase the refractive index are divided into resonant and non-resonant routes. The resonant schemes aim to minimize the factor in the denominator, $\omega_t - \omega$, by working near a resonance ($\omega_t \approx \omega$), whether it is an atomic transition level[2] or an electromagnetic resonance of artificially designed microscopic structures ('meta-atoms')[3–5]. In actual systems, the dielectric constant does not diverge on resonance due to various resonance broadening mechanisms that makes $\gamma$ a non-zero value. As one minimizes these broadening factors, the resulting index becomes larger at the design frequency; but at the same time, it becomes more frequency dispersive and the index deviates severely even for slightly different frequencies. This makes propagation of a temporal pulse impossible without distortion. This narrow band nature and enhancement-bandwidth trade-off is an intrinsic property of resonance-based designs and presents a fundamental hurdle for practical implementations of those schemes. On the other hand, there was a proposal to increase the index based on quasi-static boundary conditions, which are free from this trade-off relationship and can provide nearly frequency-independent enhancement over a broad bandwidth[6–8]. In the proposed classical model, the enhancement was shown to increase to an arbitrarily large value if the spatial gap between metallic inclusions was reduced[6,7]. However, the experimentally measured values remained < 40 (refs 8,9) as several practical and theoretical constraints impose upper bounds on the enhancement. These include lateral fabrication resolution, dielectric breakdown and a more fundamental limitation, which is the breakdown of classical material models at sub-nanometre size gaps[10]. Hence, a vitally different approach is required to enhance the refractive index much beyond the current record.

Here we report refractive indices almost two orders of magnitude larger than previous values by periodic structural designs based on space-filling geometries[11]. The geometry allows its macroscopic electric displacements to become > 1,000 times larger than the mesoscopic value, which itself is already enhanced by a similar magnitude compared with a uniform dielectric medium, resulting in gigantic electric polarizability. A dielectric constant over three million and a refractive index > 1,800 were experimentally measured in microwave frequencies, and an index of 20 was numerically verified near the optical communication wavelengths for a scaled down structure. As the enhancement principle is based on quasi-static boundary conditions, almost constant enhancement occurs for many orders of magnitude of frequencies, making this design potentially suitable for applications that require very broad bandwidth as well.

## Results

**Refractive index enhancement principle.** Figure 1a shows a schematic of the proposed structure. The key components are thin and wide metallic plates that are stacked together with insulating dielectric spacer layers (I) in an alternating A–I–B–I manner. A and B metal layers are shifted with respect to each other by a half unit cell in both lateral directions, forming a uniaxial body-centered tetragonal crystal. The unit cell size should be much smaller than the wavelength if the crystal is to be considered an effective, homogeneous medium. There are many possible choices of plate shapes and lateral array configurations in addition to the square plates in a square lattice considered here, and for all configurations, the principle of dielectric constant enhancement can be understood in terms of the enhancement of the effective polarization density for a given macroscopic electric field.

We first show that the effective AC dielectric constant for $x$-($y$-)directional field, $\varepsilon_{\text{eff},x(y)}$ (or simply, $\varepsilon_{x(y)}$), can become gigantic in the following simplified two-step explanation: (1) the enhancement of the local electric field ($E_{\text{loc}}$) over the macroscopic electric field ($E_{\text{eff}}$) due to field localization, and (2) the enhancement of the effective displacement ($D_{\text{eff}}$) over the local displacement ($D_{\text{loc}}$) due to a space-filling geometry (alternative derivation of $\varepsilon_{\text{eff}}$ using the concept of an effective capacitance can be found in Supplementary Note 1 and Supplementary Fig. 1). Here the effective field, $E_{\text{eff}}$ ($D_{\text{eff}}$), refers to the macroscopically defined electric field (electric displacement) that is uniform on the unit cell scale, while the local field, $E_{\text{loc}}$ ($D_{\text{loc}}$), is the mesoscopic electric field (electric displacement) that is highly nonuniform on the unit cell scale but uniform on the atomic scale. We make several assumptions for simplicity in this explanation, some of which will later be relaxed in our rigorous analytic model. First, we imagine an $x$-polarized macroscopic plane wave propagating in the $-z$ direction inside an infinite crystal (Fig. 1a,b). We focus on a mesoscopic region composed of several unit cells of the crystal, which is still much smaller than the wavelength, and assume that the macroscopic electric field ($E_{\text{eff}}$) is uniform in this region (Fig. 1b). Due to the smallness of the unit cell size ($a \lesssim \lambda_o/40$, where $\lambda_o$ is the wavelength of interest in vacuum) and the plate thickness ($h_m <$ skin depth), we calculate the field and charge distribution within a unit cell under the quasi-static (irrotational electric field) approximation. Assuming that the permittivity of metal is high enough in magnitude, we can neglect the electric field inside the metal, as the longitudinal electric field is blocked within a nanometre in metals (the Thomas-Fermi screening length). This allows the assignment of a single electric potential value (as denoted by $V_i$ in Fig. 1c) to each metal plate, with $\Delta V = V_{+1} - V_0 = V_0 - V_{-1} = E_{\text{eff}} \cdot a/2$. We also assume that the structure is uniform in the $y$ direction (that is, an array of infinitely long metallic strips, forming a biaxial crystal), which allows a simpler two-dimensional (2D) conceptual explanation. In our rigorous analytic model (Supplementary Note 2; Supplementary Fig. 2), we generalize it to a three-dimensional uniaxial crystal with square plates made of real metals with finite and complex permittivity and explicitly consider the effect of non-zero electric fields inside metal.

The first step, $E_{\text{loc}} = M_1 \cdot E_{\text{eff}}$, where $M_1$ is a field localization factor, is well known in the fields of metamaterials and plasmonics; the local electric field near a metal tip or within a narrow gap between metals can be much larger than the averaged electric field[8,12,13]. This enhancement is related to the screening of longitudinal electric fields inside materials with a large dielectric constant, including metals. For periodic structures, the precise relationship between $E_{\text{eff}}$ and $E_{\text{loc}}$ can be obtained by means of line integration, $\int \mathbf{E}_{\text{loc}} \cdot d\mathbf{l}$, along an arbitrary path that connects two points displaced by a unit cell vector[14,15].

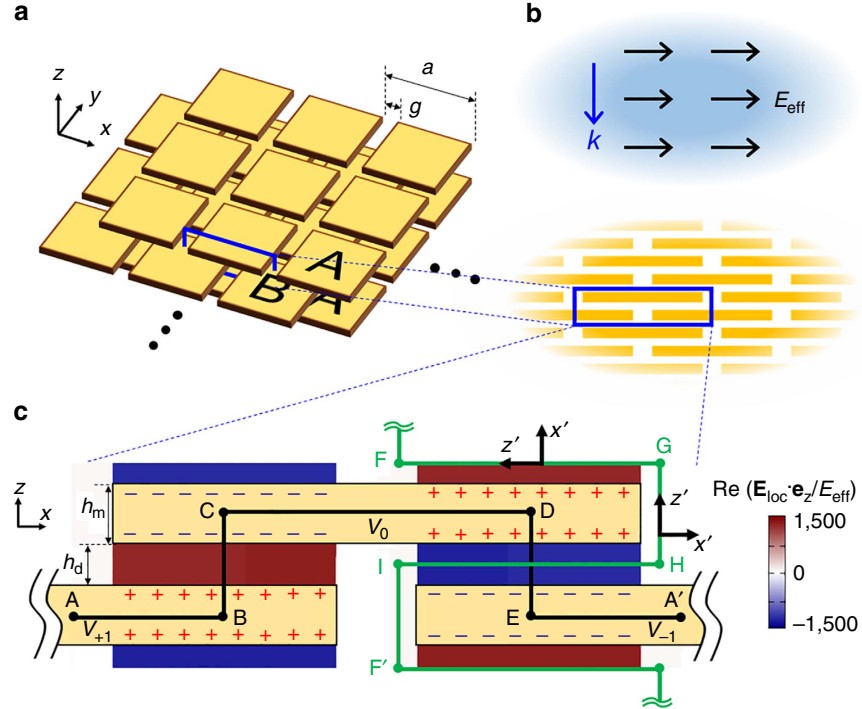

**Figure 1 | A giant refractive index mesoscopic crystal.** (**a**) A schematic of the proposed mesoscopic crystal. Electrically insulated metal plate layers are stacked in A-I-B-I fashion with half-unit cell shifts of metal layers where 'I' layers indicate insulating layers. The insulating dielectric host is not shown for clarity and fills the entire space between metals. (**b**) The mesoscopic structure (lower figure) can be homogenized to effective medium (upper figure). The blue rectangle indicates a single unit cell cross section. (**c**) The $z$ directional local electric field profile in a single unit-cell cross section was obtained from an $x$–$z$ 2-dimensional numerical simulation assuming copper in silica host (Methods). Similar profile of local electric field in $x$ direction is in Supplementary Fig. 3. The green solid line indicates the space filling curve. The structural dimensions used for the simulation were $a = 750 \, \mu m$, $g = 100 \, \mu m$, $h_m = 400 \, nm$, and $h_d = 300 \, nm$.

The A–B–C–D–E–A′ path in Fig. 1c is one example. Given that we assume the absence of an electric field inside the metal, the integrand has non-zero values only in the dielectric gaps in B–C and D–E. Inside those dielectric gaps, $\mathbf{E}_{loc}$ is nearly constant and aligned in the $\pm z$ direction because the top and bottom metallic plates effectively form a parallel-plate capacitor. Hence, the integral in each gap becomes $E_{loc} \cdot h_d$, which should be equal to $\Delta V = E_{eff} \cdot a/2$. The resulting electric field localization factor is $M_1 = a/2h_d$. The profile of $E_{loc}$ in Fig. 1c (and Supplementary Fig. 3) is directly from numerical simulations at a frequency of 5 GHz considering the actual conductivity of copper, which is in excellent agreement with the above conceptual explanation. The corresponding local electric displacement is $D_{loc} = \varepsilon_o \varepsilon_d E_{loc} = \varepsilon_o \varepsilon_d E_{eff} \cdot a/2h_d$, where $\varepsilon_d$ is the dielectric constant of the insulating dielectric host.

What is the key to the gigantic dielectric constant is the second step, $D_{eff} = M_2 \cdot D_{loc}$, where $M_2$ is the displacement enhancement factor. This results from the space-filling nature of the dielectric region and is not found in previous studies involving materials with artificial high indices. We show that the local electric displacements are rotated and accumulated to form the effective, macroscopic electric displacement. To do this, we define a local Cartesian coordinate, $(x', y', z')$, that follows the dielectric region, as depicted in Fig. 1c. As the electric displacement is the 'flux' density, the precise relationship between $D_{eff}$ and $D_{loc}$ in a crystal can be determined by performing area integration over an arbitrary surface whose boundaries lie along the corresponding paths in transversely adjacent unit cells. The solenoidal nature of the $D$ field ensures that the integration surface can be arbitrary as long as the boundaries are equal. In our 2D picture, line integration (instead of area integration) of the surface normal

flux, $\int \mathbf{D}_{loc} \cdot \mathbf{e}'_x dl$, is the relevant integration, and we selected a curve that is an equipotential contour inside the dielectric region (F–G–H–I–F′ path in Fig. 1c). This curve, at the limit of $h_d, h_m \rightarrow 0$, becomes a space-filling curve. As $(h_m + h_d) \ll a$, the integration is mostly determined by the F–G and H–I contribution, which together produces $D_{loc} \cdot a$. According to the definition of the macroscopic electric displacement, this should be equal to $D_{eff} \cdot (2h_d + 2h_m)$. Hence, $M_2 = a/2(h_d + h_m)$. We note that this factor is the ratio of the total curve length ($a$) and the straight length of the unit cell in the $z$-direction $2(h_d + h_m)$. Identical to the mathematical space-filling curves whose total length remain non-zero, while the confining area is reduced to zero[11], this factor diverges as we reduce $h_d$ and $h_m$.

One immediate observation from the derivation is that the homogenized dielectric constant is linearly proportional to $\varepsilon_d$ with a coefficient of $M_1 \cdot M_2 = a^2 [4h_d(h_d + h_m)]^{-1}$. This means that this mesoscopic structure works as a universal dielectric constant multiplier, because the enhancement coefficient is solely determined by geometric parameters and is independent of $\varepsilon_d$ and frequency. This makes the enhancement inherently a very broadband phenomenon with nearly constant enhancement from zero frequency up to the functional frequency. This statement remains true as long as quasi-static approximation is valid, and the potential variation inside each metal region is negligible. In terms of structural and material parameters, it translates to the conditions that unit cell dimension, $a$, should be much smaller than the wavelength ($a \ll \lambda_o$), and that the relative permittivity of metal, $\varepsilon_m$, is large in magnitude, satisfying $a^2/h_m h_d \ll |\varepsilon_m|/\varepsilon_d$.

Another observation is that the total enhancement factor, $M_1 \cdot M_2$, is proportional to the $(a/h)^2$, assuming $h_d = h_m = h$. Especially, the enhancement factor $M_2$, which originates from the

effective displacement manipulation, is not found in previous approaches to artificial high index materials[7,8]. This reason limited the previously measured effective dielectric constants to moderate values of a few thousands, even with a very high aspect ratio near 1,000 (ref. 8). This is several orders of magnitude below the attainable dielectric constants with the mesoscopic crystals proposed in this work with a similar aspect ratio (Supplementary Fig. 4). More fundamentally, the proposed structure for the first time shows that the macroscopic electric displacement can be markedly different from its mesoscopic vector fields, both in magnitude and in direction. A space-filling geometry is an example that utilizes this new possibility to produce gigantic macroscopic displacement from smaller local displacements with alternating directions.

**Fabrication and measurements at microwave frequencies**. We fabricated the uniaxial version of the proposed structure with square plates for the microwave operation around 10 GHz (Fig. 2a,b and Methods) for the proof-of-concept experiment. The structure can be scaled down to terahertz and visible frequency operation as well, because the geometry is simple and potentially easy to adjust to different scales. The complex effective refractive index ($n_x = \varepsilon_x^{1/2}$) was retrieved from the scattering parameters measured using the waveguide method with a vector network analyzer (Methods). As the thickness independency of effective refractive index was confirmed with numerical simulations (Supplementary Fig. 5), we used samples that are two-unit-cell thick with five metal layers for the measurements to extract the bulk properties. The retrieved dielectric constants and refractive indices were compared with quantitative theoretical predictions and to numerical simulation results. For the theoretical values, we developed an analytic model for the uniaxial mesoscopic crystal fully taking into account the finite and complex permittivity of metal (Supplementary Note 2). For the numerical simulations, we also considered the actual permittivity of metal (Methods). To demonstrate the validity of the numerically extracted bulk effective material properties, a simulation showing the Snell's law for the proposed structure is also provided (Supplementary Fig. 6).

Figure 2c,d reveal that measured effective dielectric constants and refractive indices of a mesoscopic crystal, with lateral period $a = 750\,\mu m$ and thicknesses $h_m = 400\,nm$ and $h_d = 300\,nm$, are in excellent agreement with both analytic and numerical predictions. It is noteworthy that the theoretical values and numerical results were obtained without any free parameter or fitting to the experimental results other than use of the measured conductivity

of the copper film. The measured real part of the effective dielectric constant is over $1.4 \times 10^6$ and the real part of the refractive index is over 1,200. Furthermore, these values are nearly dispersion-less, showing almost identical values over the entire X-band. Theoretically and numerically, this nearly constant trend extends down to zero frequency, which is very unusual in previous metamaterials. We note that the relative magnetic permeability was assumed to be unity in the retrieval algorithm because the thickness of metal, $h_m$, is less than the skin depth (500 to 600 nm for copper at measurement frequencies) that means the diamagnetic behaviour of metal plates is negligible. This nonmagnetic property was verified by numerical simulations that extracted the permeability, as well as the dielectric constant, without this assumption (Supplementary Fig. 7).

Figure 3 shows the dependence of the measured effective dielectric constant of the proposed structure on the aspect ratio ($a/h_d$) at 10 GHz. Again, an excellent agreement among theory, simulation, and experiment was observed. The thickness of insulating silica layer was varied ($h_d = 1,200, 600, 300$ or 150 nm), while other parameters were kept constant. For the moderate aspect ratios, the effective dielectric constant displays a quadratic dependence on the aspect ratio (green dashed line) as predicted in the simple formulation with negligible field assumption inside metal. For the larger aspect ratio, the real part of the dielectric constants does not strictly follow quadratic dependence and the imaginary part increases. This analytically predicted and experimentally measured behaviour is due to non-negligible electric fields inside metal plates. For the sample with $h_d = 150$ nm ($a/h_d = 5,000$), the measured effective dielectric constant and refractive index were $> 3.2 \times 10^6$ and 1,800, respectively. The experimentally measured imaginary part is larger than the analytic and numerical predictions, which is attributed to parasitic losses in the experimental set-up. Still, it is much smaller than the real part, meaning that the fabricated sample has a low loss tangent. For more loss sensitive applications, a smaller aspect ratio, for example 625, would provide a very large FOM ($\mathrm{Re}[n_x]/\mathrm{Im}[n_x]$) of 59 and a refractive index of 375. This is one order of magnitude larger than the previous broadband index record.

**Scaling to optical frequencies**. The principle of electric displacement manipulation with mesoscopic space-filling geometry, as well as the rigorous analytical model, can be applied to optical frequencies as well. However, because of the decrease of $\lambda$ and $|\varepsilon_m|$ for higher frequencies, the conditions mentioned previously as necessary for achieving dispersion-free, very high

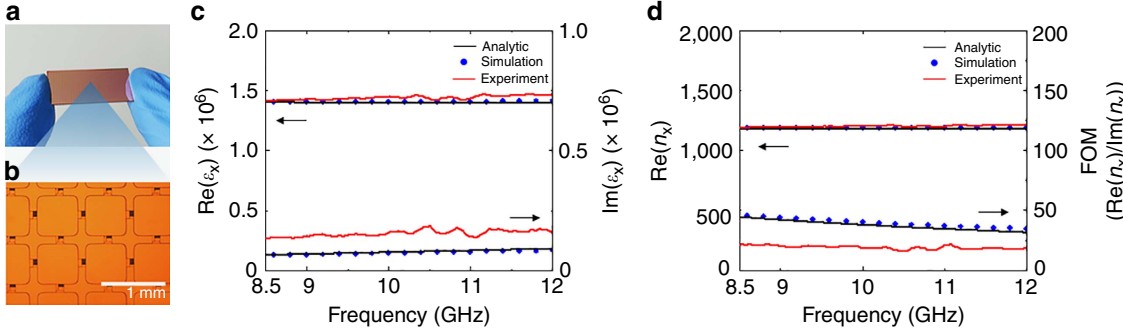

**Figure 2 | Effective material parameters of the mesoscopic crystal.** (**a**) A macroscopic image of the fabricated mesoscopic crystal. (**b**) Top view of the crystal in an optical microscope. The structural dimensions and the materials are same as in Fig. 1c, but here it is a uniaxial crystal (Methods). (**c**) Retrieved dielectric constants, $\varepsilon_x$, of the mesoscopic crystal for X-band frequency range. The analytic model, simulations, and experimental data show excellent agreement for $\mathrm{Re}[\varepsilon_x]$. (**d**) Retrieved effective refractive index and the figure of merit (FOM, $\mathrm{Re}[n_x]/\mathrm{Im}[n_x]$) of the mesoscopic crystal. The effective refractive index is constantly larger than 1000, and the experimentally measured FOM is >15 for the X-band frequency range.

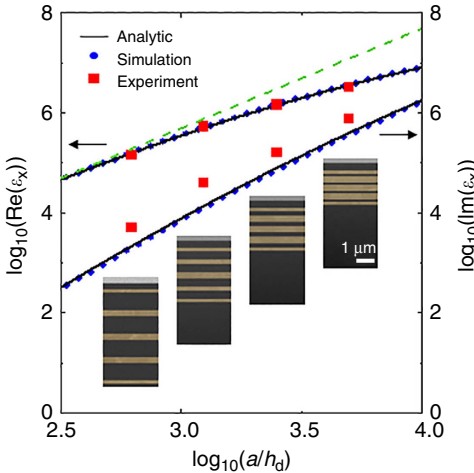

**Figure 3 | Effective dielectric constants as a function of the aspect ratio ($a/h_d$).** The homogenized dielectric constants are plotted on a logarithmic scale at 10 GHz. The green dashed line is a visual guide for quadratic dependence. The dielectric thickness of four fabricated samples are $h_d = 1,200$, 600, 300, and 150 nm, respectively, and other parameters are same as Fig. 2. Even for the sample with $h_d = 150$ nm, $\mathrm{Im}[\varepsilon_x]$ is an order of magnitude less than $\mathrm{Re}[\varepsilon_x]$, with a refractive index larger than 1,800. The scanning electron microscope images in all four insets have the same scale and are recoloured to show the metal region clearly.

refractive index (large $a/h$ while $a \ll \lambda_o$ and $a^2/h_d h_m \ll |\varepsilon_m|/\varepsilon_d$) are becoming increasingly challenging to meet. As a result, the maximum attainable index has also become progressively smaller as the frequency approaches the visible regime. Still, record-high values of refractive index with reasonable FOM can be obtained at these frequencies, as shown in Fig. 4.

In plotting Fig. 4, we used the known permittivity data for aluminium[16] for the metal and the refractive index of 1.4 for the dielectric. The structural parameters $g = h_d = h_m = 5$ nm were assumed, while $a$ was varied. In Fig. 4a, as an example, the effective indices extracted from the FDTD simulations and those calculated from the analytic model are compared for wavelengths from 0.3 to 4 μm for the fixed aspect ratio of 15 (that is, $a = 45$ nm). As in the microwave calculations, the effective index from the analytic model shows good agreement with the value retrieved from the simulation. The real part of the refractive index is $> 9$, with low dispersion and low imaginary part for wavelengths $> 2$ μm. For shorter wavelengths, both the real part and the imaginary part increase, with a resonance near 1.2 μm. This frequency dispersion is observed because the criterion for the dispersion-free high index, $a^2/h_m h_d \ll |\varepsilon_m|/\varepsilon_d$, is not fulfilled due to the smaller $|\varepsilon_m|$ for higher frequencies. If the aspect ratio is modified, a larger (smaller) refractive index with smaller (larger) FOM can be obtained. Fig. 4b,c show the analytically calculated effective index and FOM, respectively, as functions of the wavelength and aspect ratio. It is noteworthy that there exists an upper bound on the useful aspect ratio at each frequency and that this upper bound decreases as the frequency increases; this behaviour is expected because of the decreasing $|\varepsilon_m|/\varepsilon_d$. Figure 4d is a composite graph derived from the data in Fig. 4b,c demonstrating what the maximum value of the refractive index is for a given FOM as a function of wavelength. Although the structural and material conditions for the large effective index become more stringent at shorter wavelengths, the effective index can be as high as 15, with an FOM of 5, at the 1.55 μm optical communication wavelength and with realistic physical dimensions of the mesoscopic crystal. For the near infrared and

visible frequencies, the proposed structure, made of silver instead of aluminium, exhibits higher values of both maximum achievable effective index and FOM, but with a larger frequency dispersion because silver has a lower optical loss (Supplementary Fig. 8). The high index can be used for deep-subwavelength focusing and imaging. In FDTD simulations, an optical beam with a deep subwavelength waist was found, with a full-width-at-half-maximum value of $\lambda_o/23.6$ at $\lambda_o = 1.55$ μm with a plane-wave illumination on a convex lens made of the mesoscopic crystal (Supplementary Fig. 9).

## Discussion

If the designed mesoscopic crystal has a fourfold or threefold rotational symmetry axis parallel to the $z$ axis, such as in a square array of squares or a triangular array of hexagons, it becomes an optically uniaxial crystal with extreme anisotropy ($\mathrm{Re}[\varepsilon_{x(y)}]/\mathrm{Re}[\varepsilon_z] \sim 10^5$ for microwave frequencies). While an isotropic version can also be designed by adding vertical connections and reducing the aspect ratio, the extreme anisotropy of the current design is naturally ideal for applications involving deep sub-wavelength resolution image transfer (Supplementary Fig. 10), similar to the case of artificial media with hyperbolic dispersion (also known as wire media or singular media for microwave and lower frequencies)[17–20]. Although the image transfer capabilities of the proposed structure and of the hyperbolic media appear similar, there exist fundamental differences in how waves propagate in these materials. Since both $\mathrm{Re}[\varepsilon_{x(y)}]$ and $\mathrm{Re}[\varepsilon_z]$ are positive for the proposed structure, electromagnetic waves can propagate in any direction in three dimensions; the corresponding equi-frequency surface is a prolate spheroid. By contrast, the equi-frequency surface of hyperbolic media is a hyperboloid, and waves cannot propagate in directions perpendicular to the metal alignment direction (Supplementary Fig. 11). Moreover, for the allowed propagation directions, the frequency dispersion of the proposed structure are much smaller than those of the wire media (Supplementary Fig. 11). Therefore, although hyperbolic media have been proposed for many applications including super-resolution imaging and density of state enhancement[17,19], the extreme and ellipsoidal anisotropy of the proposed structure can provide unique opportunities. In addition, previously reported ellipsoidal anisotropic artificial media[21–24] have maximum index $< 15$, and their maximum index is fundamentally limited by nano-gap field enhancement, as in refs 8,9.

In summary, we have proposed and experimentally verified a mega-dielectric mesoscopic crystal with an experimentally measured real part of the refractive index of 1,800 and a dielectric constant of $3.3 \times 10^6$ in microwave frequency. The principle is based on quasi-static boundary conditions and space-filling geometries that allow inherently frequency-independent manipulation of electric displacement fields. The experimental results show excellent agreement with theoretical and numerical predictions. Scalability of the design to higher frequencies was also investigated and a $\lambda_o/23.6$ focal spot at $\lambda_o = 1.55$ μm was numerically demonstrated. The implications of the broadband extreme index and dielectric constants may extend beyond pure scientific interest to deep-sub-wavelength imaging[25], energy applications[26] and other areas in which large optical density of states over broad frequency range is important[22].

## Methods

**Fabrication of the sample.** For the X-band waveguide measurement, diced quartz wafers (0.5 mm thick) were used as substrate. The metal plates were deposited by DC magnetron sputtering of a copper target with an INVAR shadow mask. The dielectric layers were deposited by RF magnetron sputtering of a SiO$_2$ target without any mask. Lateral positioning of additional metal layers was controlled by

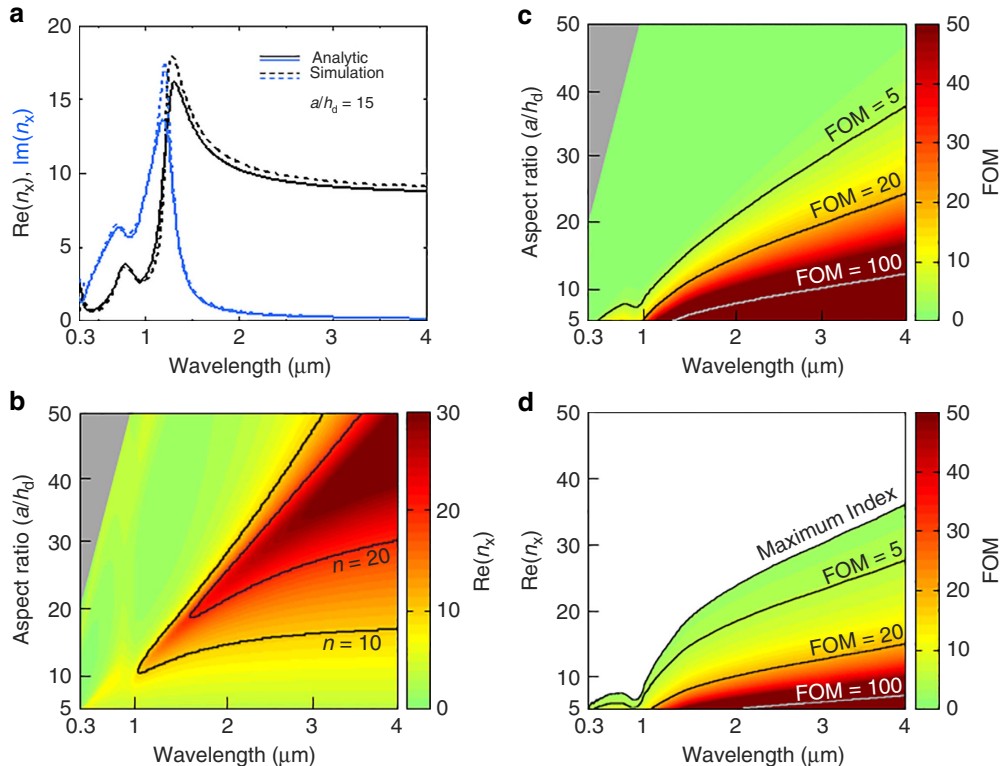

**Figure 4 | Effective parameters of the mesocopic crystal at optical frequencies.** (**a**) Analytic calculations of the effective index of the proposed mesosopic crystal with quasi-static approximation show good agreement with the refractive index retrieved from the FDTD simulation for the wavelength range of 0.3 to 4 μm. (Black and blue colours indicate the real and imaginary part; Solid and dashed lines indicate results of the analytic calculation and simulation, respectively.) (**b,c**) Wavelength dependent effective refractive index and FOM ($Re[n_x]/Im[n_x]$) for various aspect ratios are shown. Grey coloured area indicates the region for $a > \lambda_o/3$; homogenization of mesoscopic crystal may not be valid within this region. (**d**) Achievable range of effective index is plotted as a function of wavelength. Maximum achievable effective index drops as the wavelength increases; however, an effective index close to 15 can be obtained for a wavelength of 1.55 μm with an FOM of 5.

the alignment of the shadow mask. To make a stack of exactly two unit cells in the vertical direction, the bottom and uppermost metal layers had half the thickness of other metal layers. After deposition of the final metal layer, 500 nm of $SiO_2$ was deposited as a protection layer to prevent copper oxidation.

**Microwave measurement.** An X-band waveguide (X281C, Agilent) connected with a network analyzer (8510C, Agilent) was used to measure transmission coefficients, $S_{21}$, of the samples to retrieve effective dielectric constants and refractive indices with the transfer matrix method[27–29] (Supplementary Note 3). Before each measurement, the set-up was calibrated by the standard TRL 2-port calibration method. The samples were inserted into a waveguide sample holder, and silver paste was applied to the contact boundary to prevent potential leakage of electromagnetic waves. The $S_{21}$ raw data were moving averaged using a Gaussian function with full-width-at-half-maximum of 0.1 GHz. The moving averaged $S_{21}$ data were used to retrieve effective optical parameters. The $S_{21}$ raw data are in the Supplementary Fig. 12.

**Simulation.** To calculate electric fields inside a unit cell (Fig. 1c, Supplementary Fig. 3, at 5 GHz) and to retrieve effective optical parameters (Figs 2 and 3, Supplementary Figs 5, 7 and 12), a finite element method simulation tool (COMSOL Multiphysics) was utilized. The permittivity of copper was calculated from measured DC conductivity[30] (Supplementary Note 4), and that of $SiO_2$ was assumed to be 3.9. The simulation was first conducted in an $x$–$z$ 2D biaxial case, and the results were converted to the uniaxial case by multiplying a relevant geometric factor (Supplementary Note 2). The scattering coefficients, $S_{11}$ and $S_{21}$, were obtained and used to retrieve both homogenized relative permittivity and permeability (Supplementary Fig. 7) with the proper transfer matrix method (Supplementary Note 3). For $x$–$z$ 2D microwave image transfer simulation (Supplementary Fig. 10), infrared focusing simulation (Supplementary Fig. 9) and 3D Snell's law simulation (Supplementary Fig. 6), a finite-difference time-domain simulation tool (Lumerical FDTD solution), was used.

**Data availability.** The authors declare that the data supporting the findings of this study are available within the article (and its Supplementary Information files) and are available on request.

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

## Acknowledgements

This work was supported by the Samsung Science & Technology Foundation (Grant: SSTF-BA1401-06)

## Author contributions

J.S. and Y.-H.L. conceived the idea and supervised the project. T.C. and J.S. conducted the theoretical analysis. T.C. conducted numerical simulations. T.C., J.U.K., S.K.K., and H.K. fabricated and measured microwave samples. D.K.K. supervised microwave measurement.

## Additional information

**Competing financial interests:** The authors declare no competing financial interests.

