## [Peer Review File · Nature Communications]

Reviewers' comments:

Reviewer #4 (Remarks to the Author):

The authors have addressed the comments in a partially convincing way. the manuscript is rearrange and the claims, notations and directions have been clarified. I still feel that the results shown here do not present a significant advance: such performance has been demonstrated in Microwaves, albeit at a different geometry and at optical frequencies there are only simulations that show significant index (say ~ 20) but far from 1800. And again-these are "only" simulations. On the other hand, given the high quality of the work and the enthusiasm of the other referees, I do not strongly object to publications in Nat. Comm. To summarize, it is my belief that Sci. Rep. will be more suitable but I will not pose objection to publication in this journal.

Author response to the referee comments

Reviewer #4:

The authors have addressed the comments in a partially convincing way. The manuscript is rearrange and the claims, notations and directions have been clarified. I still feel that the results shown here do not present a significant advance: such performance has been demonstrated in Microwaves, albeit at a different geometry and at optical frequencies there are only simulations that show significant index (say ~ 20) but far from 1800. And again-these are "only" simulations. On the other hand, given the high quality of the work and the enthusiasm of the others referees, I do not strongly object to publications in Nat. Comm. To summarize, it is my belief that Sci. Rep. will be more suitable but I will not pose objection to publication in this journal.

A) We thank Referee #4 for reviewing of our revised manuscript. We would like to note that the image transfer simulation (now moved to supplementary information) is not the main result of the paper and just one demonstration of the effect of gigantic positive dielectric constants and refractive indices with low loss over broad bandwidth. These experimentally verified values are orders of magnitude larger than previous records (including simulation works). The refractive index is the fundamental macroscopic electromagnetic parameter which not only governs the image resolution, but also dictates Snell's law, speed of light, electromagnetic density of states, and many other properties. Therefore, in our opinion, the proposed structure should be distinguished from previous structures including hyperbolic media. For the fabrication of scaled-down structures for optical frequencies, we also hope to see further research realizing the proposed concept experimentally, which may have diverse and interesting application opportunity.